# Data Fusion of Scanned Black and White Aerial Photographs with Multispectral Satellite Images

**Dimitris Kaimaris** [1,*]**, Petros Patias** [2] **, Giorgos Mallinis** [3] **and Charalampos Georgiadis** [4]

[1]   School of Spatial Planning and Development (Eng.), Aristotle University of Thessaloniki,
    GR-541 24 Thessaloniki, Greece

[2]   School of Rural & Surveying Engineering, Aristotle University of Thessaloniki,
    GR-541 24 Thessaloniki, Greece; patias@auth.gr

[3]   Department of Forestry and Management of the Environment and Natural Resources,
    Democritus University of Thrace, GR-68200 Orestiada, Greece; gmallin@fmenr.duth.gr

[4]   School of Civil Engineering, Aristotle University of Thessaloniki, GR-541 24 Thessaloniki, Greece;
    harrisg@civil.auth.gr

*   Correspondence: kaimaris@auth.gr; Tel.: +302310991456

**Abstract:** To date, countless satellite image fusions have been made, mainly with panchromatic spatial resolution to a multispectral image ratio of 1/4, fewer fusions with lower ratios, and relatively recently fusions with much higher spatial resolution ratios have been published. Apart from this, there is a small number of publications studying the fusion of aerial photographs with satellite images, with the year of image acquisition varying and the dates of acquisition not mentioned. In addition, in these publications, either no quantitative controls are performed on the composite images produced, or the aerial photographs are recent and colorful and only the RGB bands of the satellite images are used for data fusion purposes. The objective of this paper is the study of the addition of multispectral information from satellite images to black and white aerial photographs of the 80s decade (1980–1990) with small difference (just a few days) in their image acquisition date, the same year and season. Quantitative tests are performed in two case studies and the results are encouraging, as the accuracy of the classification of the features and objects of the Earth's surface is improved and the automatic digital extraction of their form and shape from the archived aerial photographs is now allowed. This opens up a new field of use for the black and white aerial photographs and archived multispectral satellite images of the same period in a variety of applications, such as the temporal changes of cities, forests and archaeological sites.

**Keywords:** black and white aerial photographs; multispectral satellite images; data fusion; correlation tables; classification

## 1. Introduction

Data fusion is the result of using two or more images and incorporating their content information in order for the new composite image to contain more information than can be originally captured by the sensors. Image synthesis methods and techniques are used to create a high spatial resolution image that attempts to maintain the spectral information of the lower spatial resolution original data. In the composite image, the accuracy of the geometric correction can be improved, the intertemporal changes can be better defined and in addition optimal visual interpretation and classification can be allowed. Some wider areas of application are cartography, environment, urban planning, town planning, etc. [1–8].

In general, satellite imagery providers have supplied or supply multispectral (MS) images with spatial resolution four times lower than panchromatic (PAN) (e.g., Ikonos-2 at nadir 1 m PAN, 4 m MS; QuickBird-2 at nadir 0.65 m PAN, 2.6 m MS; WorldView-4 at nadir 0.31 m PAN, 1.24 m MS). Numerous data fusion images have been used so far with these ratios (1/4) of spatial resolution [9–11]. Also, data fusion images with a smaller spatial resolution ratio (e.g., 1/3) have been created, but these images come from different providers of satellite data such as e.g., Spot 10 m PAN with Landsat TM 30 m MS [12]. Finally, relatively recently, there have been image fusion attempts with a much higher spatial resolution ratio (e.g., 1/60), also from different satellite data providers such as, for example, Eros B 0.7 m PAN with Landsat-8 30 m MS [13], WorldView-2 0.5 m PAN with Landsat-8 30 m MS [14]. For all the above-mentioned image fusion cases, different methodologies and techniques have been developed over time, which in many cases, gave the same or even better results (in the sense of maintaining the same or more of the initial spectral information) of the previous methodologies and techniques of image fusion.

The different levels at which data fusion can be done are at pixel level, at feature level and at decision level [15]. Mostly, methodologies are based on techniques applied at pixel level, from which the most important being Transformation Based Fusion (Principal Component Analysis/PCA), the Additive and Multiplicative Technique (Brovey Transform, Multiplicative Technique, Color Normalized Transformation), the Wavelet Method, the Filter Fusion Method, (High-Pass Filter Fusion Method, Smoothing Filter-based Intensity Modulation), and the Fusion Based on Interband Relation (Regression fusion, Look-up-table Fusion) [6,14,16].

There are, however, a few publications on data fusion of aerial photography with satellite images. In these publications, the acquisition of aerial photographs and satellite images vary from 2 to 4 years [17–19], and this has a result that there are smaller or larger differences in the characteristics and the objects of the earth's surface which are mapped. No acquisition dates are referenced, although this information is quite important as different capturing times add errors when composing images. In addition, either quantitative controls are not performed on these tests [17], or the aerial photographs are recent and colorful [18], or only the RGB bands are used for satellite image fusion [19]. Therefore, it can be stated that the addition of multispectral information (MS) from an satellite image to an black and white aerial photograph of the 2nd half of the 20th century (1950–1999), with a small difference (just a few days) in their acquisition date, the same year and season, has not been studied in detail up to date. A feasibility study of the effectiveness of this idea will open up a new field of use of aerial photographs and satellite images.

The geometry of images differs in aerial photographs and satellite images, whether they are recent or older. For example, black and white aerial photographs are central projections, that is, they result from the projection rays of the Earth's surface through the center of the lens onto the film plate. In the case of satellite images, e.g., of Landsat 5, where each pixel of the image is gradually captured by the rotation of the sensor mirror, the image is created by thousands of central projections. Besides this fact, the altitude of acquisition in aerial photographs and satellite images varies by hundreds of kilometers. Also, black and white aerial photographs do not provide any spectral information beyond the visible, but have a particularly high spatial resolution. As a result, it is not possible to apply classification algorithms to the image and, as a consequence, it is impossible to automatically extract the land covers of the Earth's surface. Instead, numerous satellite image files (such as Landsat 5) have channels with a variety of spectral information, but they are accompanied by bad spatial resolution. In these images, classification can be performed, but, due to poor spatial resolution, pixel-level generalizations are large. Therefore, the question that arises refers to the possibility of re-using archived black and white aerial photographs after being improved by using the spectral information of archived satellite images. Obviously, it is important that the acquisition dates are identical or differ for just a few days and that they are acquired in the same year. Thus, the fusion of these archived data and the results of their classifications are the objectives of this paper.

## 2. Data

Black and white (B/W) aerial photographs (film of high sensitivity of 250 lp/mm) of 1987 and 1990 of Sparta and Pyrgos area (Figures 1 and 2) were used respectively, at a scale of 1:20,000 (Table 1), which were scanned at 1200 dpi. In both areas, urban and sub-urban areas are included in aerial photographs, while the selection of locations was the result of the first searches of the editorial team for data with small difference in their acquisition date (just a few days), the same year and season. In each study area the aerial photographs belong to one strip and have an overlap of 60%. All aerial photographs were accompanied by camera calibration data (Camera Calibration RMK A 15/23 and Camera Calibration RMK A). Moreover, atmospheric corrected satellite images Landsat 5 of 1987 (Figure 3) and 1990 were collected in the respective geographic areas (U.S. Geological Survey [20]). Aerial photographs of 1987 were captured 7 days before, while 1990 aerial photographs were taken 1 day after the satellite images were captured in the above geographic areas. Finally, for the realization of the aerial triangulation, Ground Control Points (GCPs) and Check Points (CPs) were used, whose horizontal coordinates were collected from the Hellenic Cadaster [21], i.e., the official public provider of cartographic base maps in Greece, with horizontal accuracy of 1 m. The corresponding altitude information was collected from the DTM (5 × 5 m point grid, 1.5 m vertical accuracy) also from the Hellenic Cadastre.

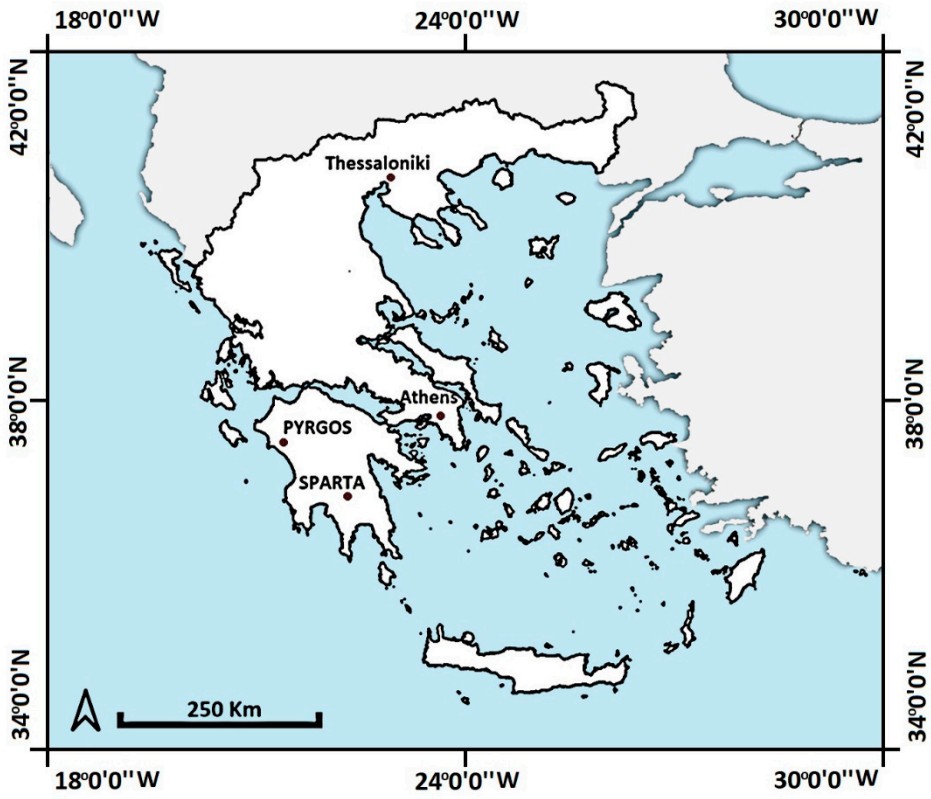

**Figure 1.** Map of Greece with the location of Sparta and Pyrgos.

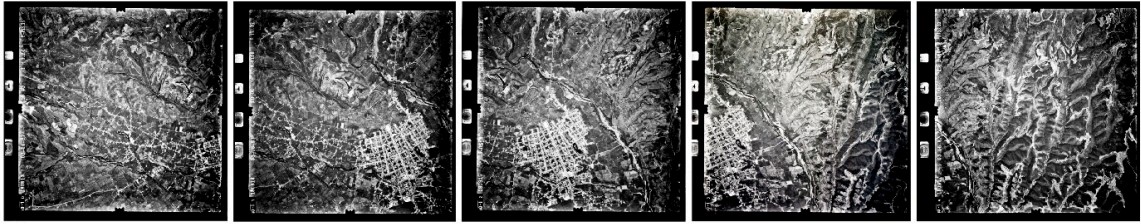

**Figure 2.** The case of aerial photographs of the Spartan area. Flight direction left to right, west to east.

**Table 1.** Characteristics of Aerial photographs and satellite images.

| Data | Location | Number of Images | Date of Capture | Spectral Resolution | Spatial Resolution | Radiometric Resolution |
|---|---|---|---|---|---|---|
| Aerial photographs | Sparta | 5 | 03/06/1987 | b/w, visible spectrum | 0.50 m | 8 bit |
| | Pyrgos | 5 | 29/08/1990 | | 0.50 m | |
| Satellite images Landsat 5 | Sparta | 1 | 10/06/1987 | 6 Bands: R-G-B-NIR-SWIR1-SWIR2 | 30 m | |
| | Pyrgos | 1 | 28/08/1990 | | | |

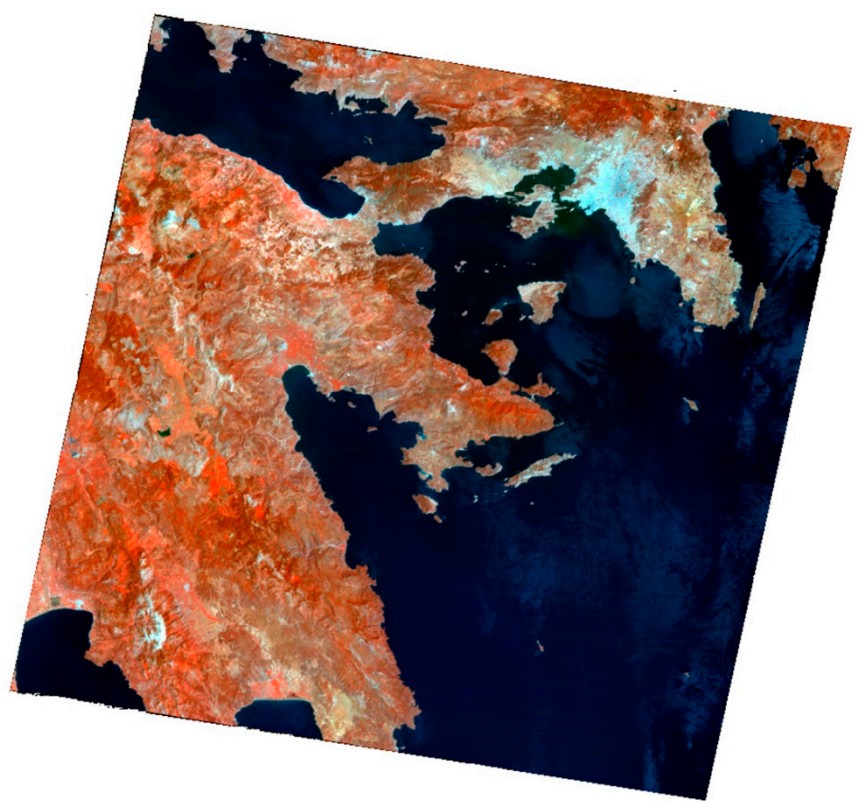

**Figure 3.** The case of the satellite image Landsat 5, 10/06/1987, 30 m, B-G-NIR.

## 3. Methodology and Processing of Data/Products

After collecting the necessary data, the aerial triangulation-s of aerial photographs (for the production of orthophoto mosaics) and geometric correction-s of satellite images are conducted in the study area. Following are the image fusions of the orthophoto mosaics of the aerial photographs with the Landsat 5 orthoimages, as well as the checking of the quality (correlation tables) of the fused spectral information of the Multispectral (MS) satellite. Finally, the following classifications, both on the composite as well as on the MS ortho images, will be checked for their capability to provide accurate area measurements of both built and open surfaces.

### 3.1. Aerial Triangulation

Aerial triangulation was performed with Leica Photogrammetry Suite (LPS) of Erdas Imagine©. Using the Camera Calibration files, the interior orientations of the aerial photographs were restored, 13 GCPs and 5 CPs were selected in the two study areas (Figures 4a and 5a). Digital Surface Model (DSM) of 5 m spatial resolution and the orthophoto mosaics (Figures 4b and 5b) of the study area with spatial resolution of 0.5 m were produced.

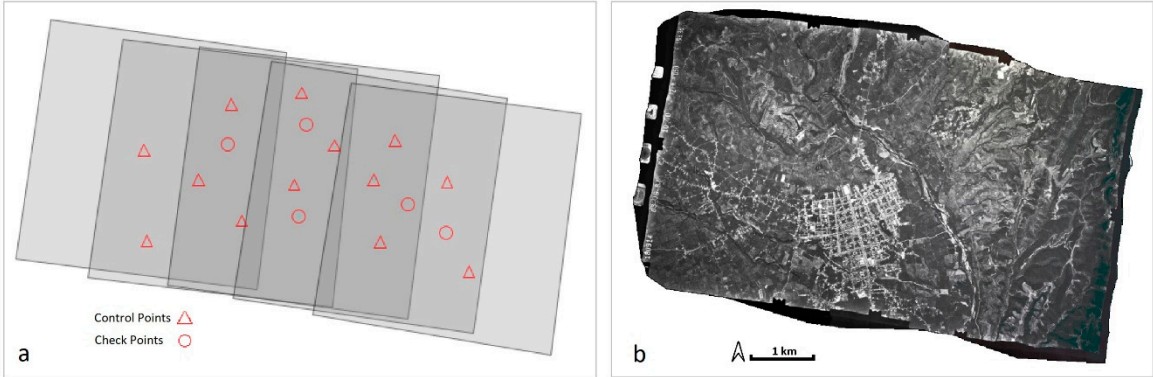

**Figure 4.** (**a**) The distribution of GCPs and CPs in the wider area of the city of Sparta. (**b**) The orthophoto mosaic of aerial photographs.

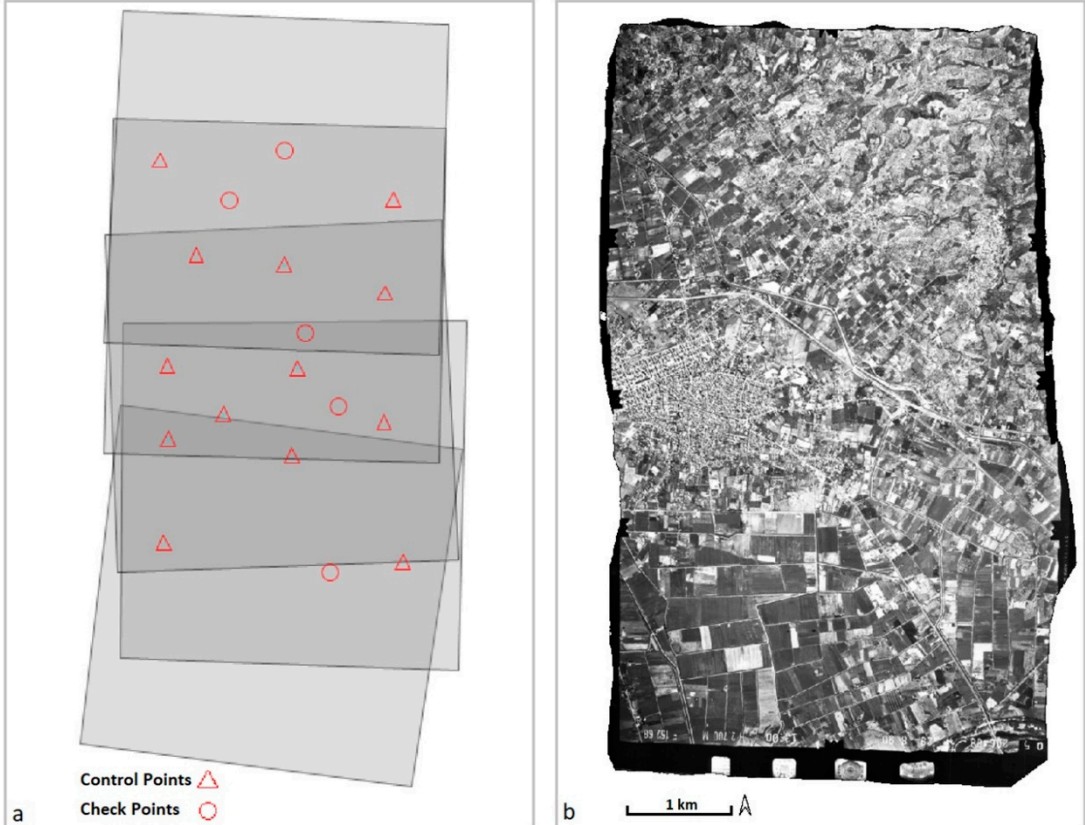

**Figure 5.** (**a**) The DSM of the wider area of the city of Pyrgos. (**b**) The orthophoto mosaic of aerial photographs.

After the completion of the aerial triangulations, the CPs were used to calculate the estimates of the differences $\hat{\Delta}x$ and $\hat{\Delta}y$ and the standard deviations $\hat{\sigma}_X$ and $\hat{\sigma}_Y$ (Table 2).

**Table 2.** Spatial accuracy test of the geometric-corrected image (measurement units: meters).

| Estimated Indices (units m) | Orthophoto Mosaic from Aerial Photographs | | Orthoimagery from Satellite Images | |
| --- | --- | --- | --- | --- |
| | Study Areas | | | |
| | Sparta | Pyrgos | Sparta | Pyrgos |
| $\hat{\Delta}x = \frac{\sum_{i=1}^{n} \delta x_i}{n} = \frac{\sum_{i=1}^{n} \lvert x_{ORTHO,i} - x_{CP,i}\rvert}{n}$, where $\delta x_i$ the differenced of CPs in the X axis between the orthoimage and the actual values, $x_{ORTHO,i}$ the values of CPs in the X axis in the orthoimage, $x_{CP,i}$ the actual values of CPs in the X axis, and $n$ the number of observations (=5). | 2.1 | 1.0 | 9.5 | 7.2 |
| $\hat{\Delta}y = \frac{\sum_{i=1}^{n} \delta y_i}{n} = \frac{\sum_{i=1}^{n} \lvert y_{ORTHO,i} - y_{CP,i}\rvert}{n}$ | 2.4 | 1.0 | 9.2 | 8.3 |
| $\hat{\sigma}_X = \sqrt{\frac{1}{n-1}\sum_{i=1}^{n}\left(\delta x_i - \hat{\Delta}x\right)^2} = \sqrt{\frac{1}{n-1}\sum_{i=1}^{n}\left(\lvert x_{ORTHO,i} - x_{CP,i}\rvert\right) - \hat{\Delta}x)^2}$ | 1.5 | 0.4 | 3.6 | 5.7 |
| $\hat{\sigma}_Y = \sqrt{\frac{1}{n-1}\sum_{i=1}^{n}\left(\delta y_i - \hat{\Delta}y\right)^2} = \sqrt{\frac{1}{n-1}\sum_{i=1}^{n}\left(\lvert y_{ORTHO,i} - y_{CP,i}\rvert\right) - \hat{\Delta}y)^2}$ | 1.8 | 0.3 | 4.2 | 4.8 |

## 3.2. Geometric Correction of Satellite Images

For the collection of GCPs, in order to perform the geometric correction of satellite images (image resection using Erdas Imagine©), the products of the aerial photography were utilized, namely the orthophoto mosaics and DSMs, were used. Substantially the image (of the satellite) was recorded on another image (orthophoto mosaic of aerial photographs). In each study area, 10 GCPs and 5 CPs were used. After the geometric corrections, the CPs were used to calculate the estimates of the differences $\hat{\Delta}x$ and $\hat{\Delta}y$ and the estimates of the standard deviations $\hat{\sigma}_X$ and $\hat{\sigma}_Y$ (Table 2).

## 3.3. Fusion of Images

The fusion of Orthophoto mosaics of the aerial photographs with the Landsat 5 orthoimages, as well as the checking of the quality of the fused spectral information of the Multispectral (MS) satellite, was carried out in Erdas Imagine©. Figures 6 and 7 show the b/w orthophoto mosaics (Figures 6a and 7a) and the multi-spectral satellite images (Figures 6b and 7b) in rectangular sections in the study areas. Initially, Resolution Merge was performed using the Principal Component Analysis method, Bilinear Interpolation and Output 8 bit data reconstruction technique, producing a data-fusion image for each study area (Figure 6c,d and Figure 7c,d). The evaluation of fused image is based on qualitative-visual analysis and quantitative-statistical analysis. The qualitative-visual analysis is subjective and is directly related to the experience of the fused image creator (e.g., are more details recognized in the image or are colors, contrasts preserved? etc.) [7]. The quantitative-statistical analysis is objective and is based on spectral analysis of images. The most commonly used method is the correlation coefficient between the original bands of the MS image and the corresponding bands of the fused image. The correlation coefficient values range from −1 to 1. Usually, the values between the corresponding bands of the two images (MS and fused image) must be from 0.9 to 1, so that the fused image can be used for the e.g., successful classification of earth's surface coverings and objects [7,22–24]. In order to create the correlation tables (Tables 3 and 4) of ortho multispectral satellite images with composite images, composite images were degraded spatially (60 times, image degradation) to obtain the spatial resolution of MS images.

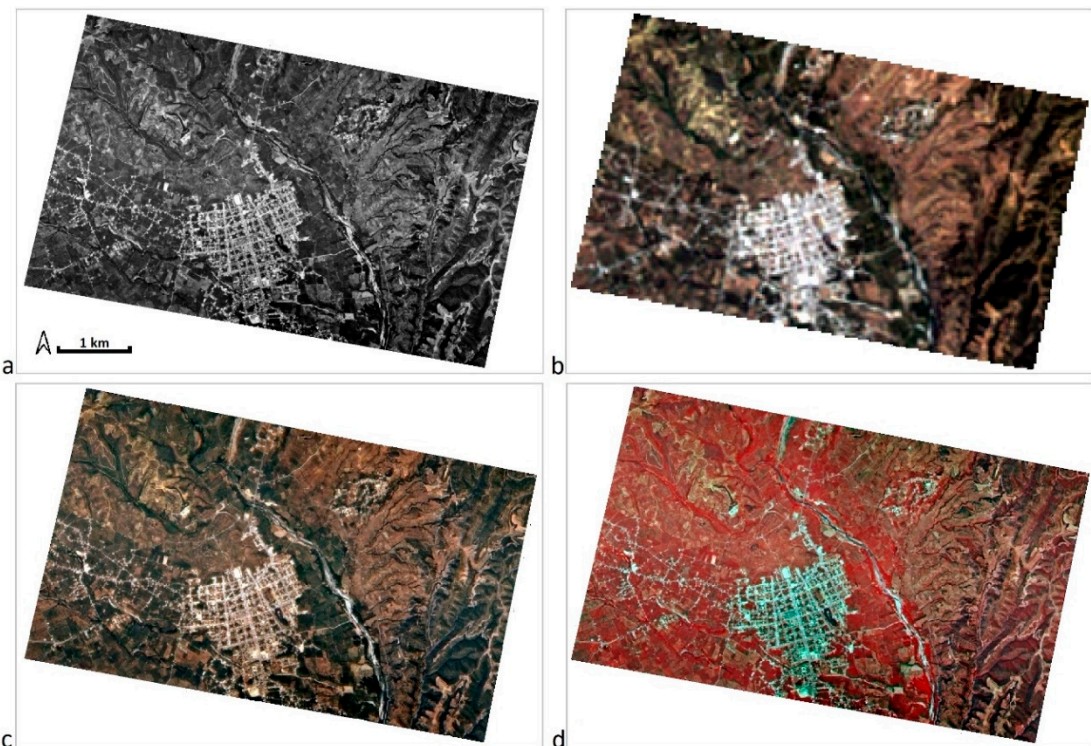

**Figure 6.** Wider area of Sparta. (**a**) Cutting the orthophoto mosaic of the aerial photographic into a rectangle. (**b**) The corresponding area of the Landsat 5 orthoimage. (**c**) The composite image (B-G-R). (**d**) The composite image (B-G-NIR).

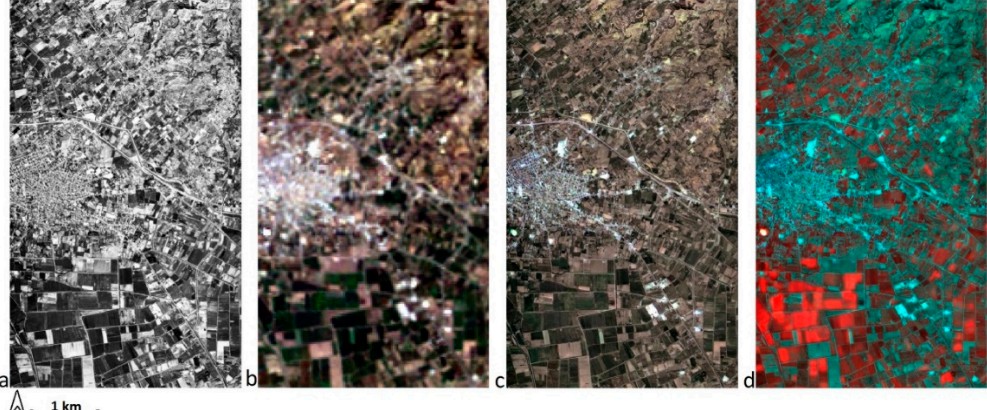

**Figure 7.** Wider area of Pyrgos. (**a**) Cutting the orthophoto mosaic of the aerial photographic into a rectangle. (**b**) The corresponding area of the Landsat 5 orthoimage. (**c**) The composite image (B-G-R). (**d**) The composite image (B-G-NIR).

**Table 3.** Correlation table of the satellite multispectral orthoimages with the spatially degraded composite image in Spartan city area.

| | Bands | LANDSAT 5 | | | | | | DATAFUSION | | | | | |
|---|---|---|---|---|---|---|---|---|---|---|---|---|---|
| | | Blue | Green | Red | NIR | SWIR1 | SWIR2 | Blue | Green | Red | NIR | SWIR1 | SWIR2 |
| **LANDSAT 5** | Blue | 1 | 0.979 | 0.927 | 0.203 | 0.751 | 0.883 | *0.697* | 0.750 | 0.774 | 0.193 | 0.584 | 0.775 |
| | Green | 0.979 | 1 | 0.969 | 0.239 | 0.827 | 0.925 | 0.693 | *0.770* | 0.815 | 0.240 | 0.656 | 0.815 |
| | Red | 0.927 | 0.969 | 1 | 0.204 | 0.894 | 0.943 | 0.658 | 0.751 | *0.843* | 0.239 | 0.721 | 0.833 |
| | NIR | 0.203 | 0.239 | 0.204 | 1 | 0.373 | 0.154 | −0.010 | 0.048 | 0.068 | *0.528* | 0.146 | 0.028 |
| | SWIR1 | 0.751 | 0.827 | 0.894 | 0.373 | 1 | 0.914 | 0.467 | 0.581 | 0.705 | 0.297 | *0.735* | 0.742 |
| | SWIR2 | 0.883 | 0.925 | 0.943 | 0.154 | 0.914 | 1 | 0.612 | 0.701 | 0.781 | 0.179 | 0.707 | *0.849* |
| **DATAFUSION** | Blue | *0.697* | 0.693 | 0.658 | −0.010 | 0.467 | 0.612 | 1 | 0.978 | 0.909 | 0.564 | 0.803 | 0.878 |
| | Green | 0.750 | *0.770* | 0.751 | 0.048 | 0.581 | 0.701 | 0.978 | 1 | 0.964 | 0.565 | 0.866 | 0.934 |
| | Red | 0.774 | 0.815 | *0.843* | 0.068 | 0.705 | 0.781 | 0.909 | 0.964 | 1 | 0.505 | 0.914 | 0.963 |
| | NIR | 0.193 | 0.240 | 0.239 | *0.528* | 0.297 | 0.179 | 0.564 | 0.565 | 0.505 | 1 | 0.656 | 0.442 |
| | SWIR1 | 0.584 | 0.656 | 0.721 | 0.146 | *0.735* | 0.707 | 0.803 | 0.866 | 0.914 | 0.656 | 1 | 0.913 |
| | SWIR2 | 0.775 | 0.815 | 0.833 | 0.028 | 0.742 | *0.849* | 0.878 | 0.934 | 0.963 | 0.442 | 0.913 | 1 |

**Table 4.** Correlation table of the satellite multispectral ortho multispectral satellite images with the spatially degraded composite image in Pyrgos city area.

| | Bands | LANDSAT 5 | | | | | | DATAFUSION | | | | | |
|---|---|---|---|---|---|---|---|---|---|---|---|---|---|
| | | Blue | Green | Red | NIR | SWIR1 | SWIR2 | Blue | Green | Red | NIR | SWIR1 | SWIR2 |
| **LANDSAT 5** | Blue | 1 | 0.967 | 0.947 | −0.153 | 0.707 | 0.832 | *0.811* | 0.773 | 0.726 | −0.290 | 0.553 | 0.595 |
| | Green | 0.967 | 1 | 0.963 | −0.092 | 0.738 | 0.855 | 0.800 | *0.818* | 0.756 | −0.255 | 0.582 | 0.622 |
| | Red | 0.947 | 0.963 | 1 | −0.244 | 0.818 | 0.921 | 0.792 | 0.786 | *0.789* | −0.399 | 0.668 | 0.702 |
| | NIR | −0.153 | −0.092 | −0.244 | 1 | −0.106 | −0.260 | −0.179 | −0.110 | −0.225 | *0.870* | −0.228 | −0.288 |
| | SWIR1 | 0.707 | 0.738 | 0.818 | −0.106 | 1 | 0.926 | 0.568 | 0.580 | 0.639 | −0.279 | *0.737* | 0.681 |
| | SWIR2 | 0.832 | 0.855 | 0.921 | −0.260 | 0.926 | 1 | 0.676 | 0.681 | 0.716 | −0.402 | 0.715 | *0.736* |
| **DATAFUSION** | Blue | *0.811* | 0.800 | 0.792 | −0.179 | 0.568 | 0.676 | 1 | 0.952 | 0.947 | −0.421 | 0.757 | 0.809 |
| | Green | 0.773 | *0.818* | 0.786 | −0.110 | 0.580 | 0.681 | 0.952 | 1 | 0.948 | −0.348 | 0.743 | 0.793 |
| | Red | 0.726 | 0.756 | *0.789* | −0.225 | 0.639 | 0.716 | 0.947 | 0.948 | 1 | −0.500 | 0.872 | 0.910 |
| | NIR | −0.290 | −0.255 | −0.399 | *0.870* | −0.279 | −0.402 | −0.421 | −0.348 | −0.500 | 1 | −0.524 | −0.569 |
| | SWIR1 | 0.553 | 0.582 | 0.668 | −0.228 | *0.737* | 0.715 | 0.757 | 0.743 | 0.872 | −0.524 | 1 | 0.938 |
| | SWIR2 | 0.595 | 0.622 | 0.702 | −0.288 | 0.681 | *0.736* | 0.809 | 0.793 | 0.910 | −0.569 | 0.938 | 1 |

### 3.4. Classifications and Area Measurements

The classifications that follow, both on the composite as well as on the MS ortho images, will be checked for the capability of providing correct area measurements of both built and open surfaces, into smaller geographical areas (Figure 6c compared with Figures 8a and 7c with Figure 9a). The use of smaller geographic areas will avoid generalizations more apparent in geographically larger areas. As reference data for and comparison reasons, the results of photo interpretation and manual rendering of complex images will be used.

For the clarification of composite and MS satellite orthoimages [25], the Unsupervised -a Pixel-based technique- and ISODATA as a classifier were used. 35 classes (by estimation) of classification were selected, then grouped into regions of built and open surface (Figure 8c,d and Figure 9c,d) and, finally, their areas were automatically calculated (Table 5). The above processing was performed with Erdas Imagine©.

In order to determine the actual areas of the built and open surface, composite images were introduced into ArcGIS©, where the digitization (Figures 8e and 9e) and the automated calculations of their areas (Table 5) were performed.

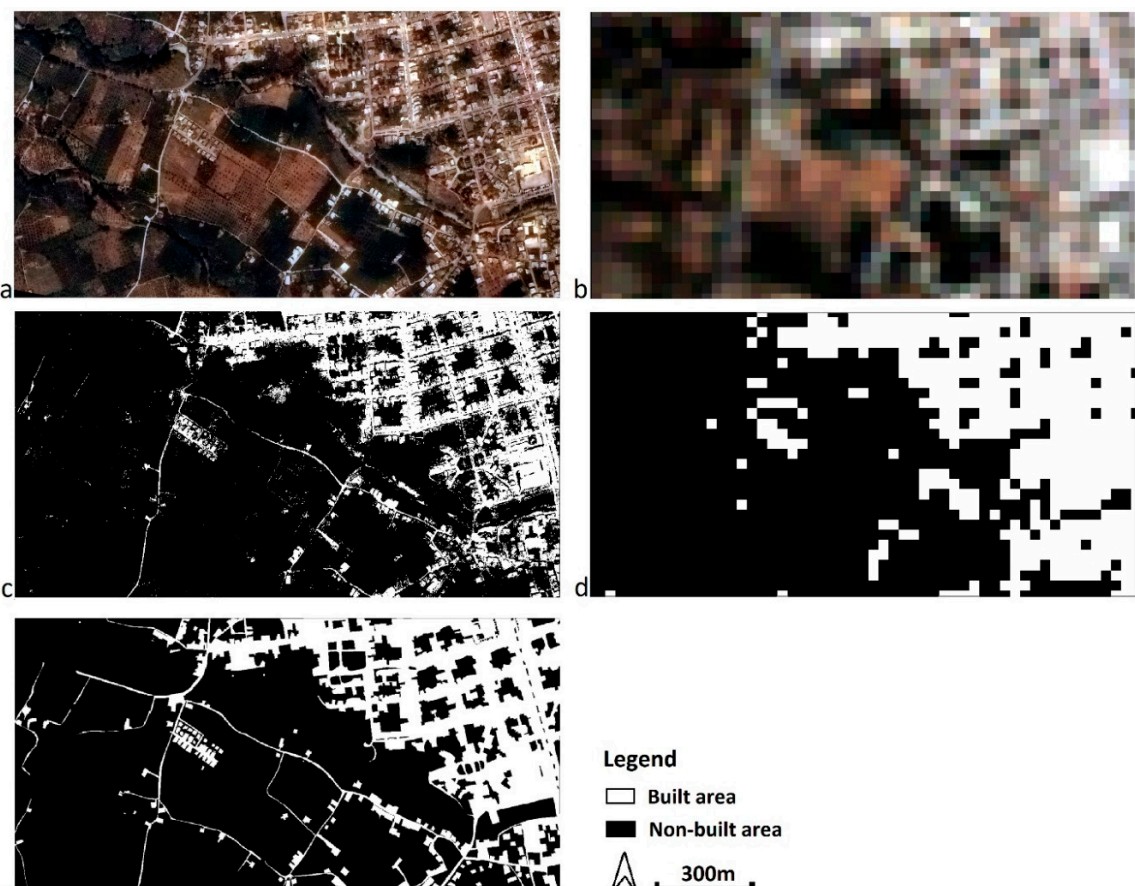

**Figure 8.** (**a**) Part of the original composite image (B-G-R) of Sparta's wider area. It includes a built and open surface, urban and agricultural. (**b**) The corresponding part of the orthoimage MS Landsat 5 (B-G-R). (**c**) Classification of the composite image. (**d**) Classification of the orthoimage MS Landsat 5. (**e**) Digitization of the composite image in ArcGIS©.

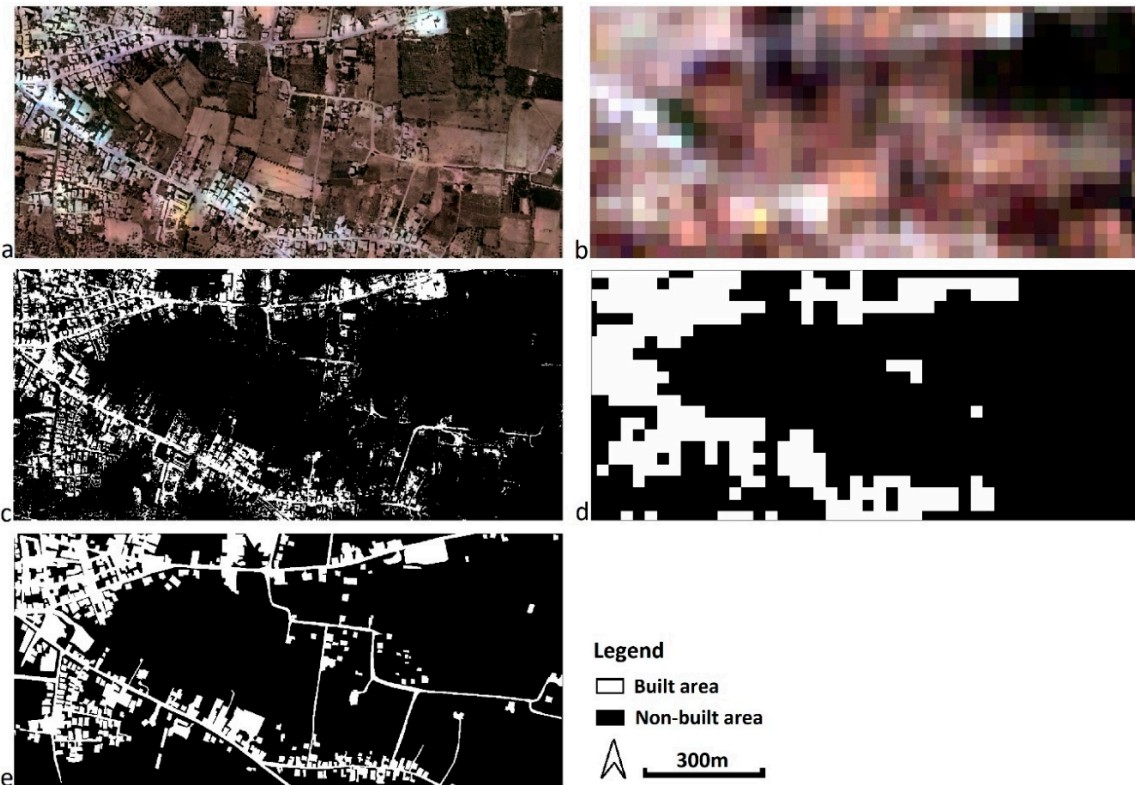

**Figure 9.** (**a**) Part of the original composite image (B-G-R) of Pyrgos wider area. It includes a built and open surface, urban and agricultural. (**b**) The corresponding part of the orthoimage MS Landsat 5 (B-G-R). (**c**) Classification of the composite image. (**d**) Classification of the orthoimage MS Landsat 5. (**e**) Digitization of the composite image in ArcGIS©.

**Table 5.** Comparison of the built and open surface resulting between the comparison of the GIS digitization with the classification of the images.

|  |  | Digitization in GIS | | Classification Landsat 5 | Classification Datafusion | |
|---|---|---|---|---|---|---|
|  |  | Area (sqm) | Area (sqm) | Difference % | Area (sqm) | Difference % |
| Sparta | Built surface | 349,332.31 | 453,600.00 | 29.80 | 274,119.00 | −21.5 |
|  | Open surface | 1,022,910.195 | 918,642.50 | −10.19 | 1,098,123.50 | 7.40 |
| Pyrgos | Built surface | 159,466.43 | 211,500.00 | 32.60 | 129,493.75 | −18.80 |
|  | Open surface | 676,364.56 | 624,330.99 | −7.70 | 706,337.24 | 4.43 |

## 4. Results

The wider areas of two Greek cities are studied in the paper, in order to check the effectiveness of the composition process of satellite images with b/w aerial photographs with small difference (just a few days) in the image acquisition date, the same year and season, in areas with diversified coverages.

The geometric accuracy of the produced orthophoto mosaic and orthoimages is presented in Table 2. In particular, the achieved results in terms of standard errors on the X and Y axis range 0.4–1.8 m (estimates of the differences 1.0–2.4 m) in the case of orthophoto mosaics of aerial photographs and 3.6–5.7 m (estimates of the differences 7.2–9.5 m) in the case of satellite orthoimages.

Satellite image providers supply MS images with a spatial resolution four times larger than the panchromatic. This was simulated by producing the orthophoto mosaics of the aerial photographs

spatially degraded from 0.5 m to 7.5 m (four times smaller than the spatial resolution of the MS image) and then the fusion of the images was carried out. From the Correlation Tables that created, it is shown that the rates of retaining the original spectral information of the ortho MS images in the composite images are similar (at least if not better) to the corresponding percentages of the non-initial degradation of the orthophoto mosaics of the aerial photographs presented in this paper. For this reason, the case of the initial degradation of orthophoto mosaics was not further analyzed in the paper.

In addition, during the fusion of the images (with or without initial degradation of the orthomosaics of aerial photographs) apart from the Principal Component Analysis Method (Transformation Based Fusion), other methods were used, such as Multiplacative and Brovey Transform (Additive and Multiplicative Technique) [6,26–34], which did not provide better results in the maintenance of spectral information, and, therefore, were not further analyzed in the paper.

According to the Correlation Table (Table 3) for the wider area of the city of Sparta, the transmission of the spectral information from the MS satellite orthoimage to the composite image is performed satisfactory by 70–85% (except for NIR that is 53%). Also, for the wider area of the city of Pyrgos, it is found that the transmission of the spectral information from the MS satellite orthoimage to the composite image is, also, performed satisfactory by 74–82%. Consequently, despite the fact that the rates are not from 90–100% (ideally), these are generally satisfactory and composite images may be used to perform further classifications.

Table 5 shows that in the case of the composite image, the overall estimate of built surfaces is about 20% underestimated, while the overall estimate of open surfaces is about 6% over-estimated. In the case of the original MS satellite images, the overall estimate of built surfaces is approximately 31% over-estimated, while the overall estimate of open surfaces is about 9% underestimated. The important conclusion is that the calculated areas of both the built and the open surfaces in the case of classification of composite images are much closer to reality. In addition, it is of major importance that in the case of composite image classification, it is possible to automatically extract the geometry/shape of the earth's objects (compare Figure 8c with Figures 8e and 9c to Figure 9d), which is not possible with the classification of the Landsat 5 satellite images (compare Figure 8d with Figures 8e and 9d with Figure 9e). The above is obviously due to the very good spatial resolution of composite images with respect to the lower spatial resolution of Landsat 5 images.

## 5. Discussion

Few publications have been made on data fusion of aerial photography with satellite images. In addition to their scarcity, the acquisition date of aerial photographs differs from the acquisition date of satellite images from 2 to 4 years. This fact clearly results in differences—small or large—in the characteristics and objects of the earth's surface that are documented, triggering difficulties and distortions during their study. Moreover, the acquisition dates are not mentioned, despite their importance, as different acquisition periods create errors in the composition of images. Lastly, three issues emerge, not necessarily at the same time: only the RGB bands are used for satellite image fusion, no additional quantitative controls are conducted and the aerial photographs are recent (21th century) and colorful.

No new technique for image composition is presented in the paper, but for the first time in international literature, the addition of multispectral information from satellite images to b/w aerial photographs with small difference in their image acquisition date (just a few days), the same year and season is presented.

In fact, in several cases of composing modern PAN and MS images of the same satellite system, the acceptable limits of retaining the original MS image spectral information in the composite image are either not satisfied or marginally satisfied. Even more so, if images come from such different sensors as in this paper, it is expected that the above limits could not be met. Therefore, it is of primary interest to determine the percentage of spectral information retained in the composite image, and whether this percentage permits the classification of new images, the digital identification of basic structures of the

terrestrial surface (built area and non-built area) and the extraction of quantitative data. The results have allowed the above to reach a satisfactory level of acceptance and, in consequence, the research on the creation of new techniques and methodologies for their improvement can now begin.

## 6. Conclusions

The paper highlights a new possibility of using archived b/w aerial photographs, since their spectral information can be improved. It is a prerequisite that the corresponding spatial multi-spectral satellite data have the same date of acquisition or differ for just a few days from the acquisition dates of the aerial photographs. New possibilities of using archived Landsat 5 satellite images are also highlighted.

Flights for b/w aerial photographs, in the past, aimed at studies on Cadastre, Forestry, Town Planning, Spatial Planning, etc. Each flight was accompanied by the acquisition of hundreds of aerial photographs, which, nowadays, will allow the enrichment with multi-spectral data of orthophoto mosaics that map spatially large areas. Therefore, automatic extraction of temporal information, e.g., of built and open areas on hundreds of square kilometers of land, will no longer be the result of continuous photointerpretation process and rendering of orthophoto mosaics, but of their automatic extraction through automatic classification. Finally, a variety of new applications is open, such as the identification of the temporal changes of cities, forests, archaeological sites, etc.

Obviously, the data fusion of high spatial resolution images with low spatial resolution ones, as in our case, is accompanied by the problem of large ratios of their spatial resolutions, which often causes the appearance of spectral distortions. However, as the first results are encouraging, the development of optimizations to minimize these spectral distortions are now possible.

**Author Contributions:** D.K. proposed the core concept of the paper. The work presented in this paper was carried out in collaboration between all authors. All authors approved the submitted manuscript. All authors contributed to the scientific content, the processing of data, the interpretation of the results and manuscript revisions. All authors have read and agreed to the published version of the manuscript.

**Funding:** This research received no external funding.

**Conflicts of Interest:** The authors declare no conflict of interest.

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
