# Peer review of "Data Fusion of Scanned Black and White Aerial Photographs with Multispectral Satellite Images"

_sci, doi:10.3390/sci2020029_

Round 1

Reviewer 1 Report

In this manuscript, the authors are presenting the fusion of b/w aerial photography with multispectral satellite (Landsat-5) images having a few days difference in the acquisition date. More specific the authors are studying the addition of multispectral information from satellite images to black and white aerial photographs for classifying build and non-build areas in two Greek cities (Sparta and Pyrgos).

The manuscript is well organized, with the introduction having adequate references and the material and methods section describing in a correct manner the methods used. Moreover, in the results and discussion section, the authors are transparently presenting their findings.
Overall the manuscript is well written, and authors were presenting in detail all the necessary information for their approach; thus, I believe that should be published in the journal.

The following comments focus on the improvements that would benefit the manuscript.
In the abstract, the authors should change their statement
" from satellite images to black and white aerial photographs of the 2nd half of the 20th century (1950–1999)" to " from satellite images to black and white aerial photographs of the 80s decade (1980–1990)" as they only used data acquired in this time frame. The statement "2nd half of the 20th century" may confuse the readers as of the numbers and the time frame of the used datasets.

In table 2, authors should consider adding measurement units (meters) to their results.

In Figures 6 and 7 caption " rectangular rectangle." = " rectangle"
In Table 3 for Landsat data fusion results in Green column and Green Line "emph0.770" = "0.770".

In the 3.4 Classification subsection, the authors' stating that they used 35 classification classes that subsequently are grouped into two classes. Which are these classes, and how they are grouped? These 35 classes are, for example, the classes that used are part of the 44 Corine Classes?

Can this methodology be applied to currently available data (new high-resolution aerial data and Sentinel imagery)?
If the answer is positive, this should be stated in the manuscript as can be a crucial conclusion for applying this data fusion methodology, for example, to the automatic extraction of temporal information in Corine classification, reducing the photointerpretation process duration.

Author Response

In the abstract, the authors should change their statement " from satellite images to black and white aerial photographs of the 2nd half of the 20th century (1950-1999)" to " from satellite images to black and white aerial photographs of the 80s decade (1980-1990)" as they only used data acquired in this time frame. The statement "2nd half of the 20th century" may confuse the readers as of the numbers and the time frame of the used datasets. We would like to thank the reviewer for his constructive comments and suggestions that have improved our paper. We agree with the reviewer and change in the Abstract (in this section only) the statement FROM "from satellite images to black and white aerial photographs of the 20th century (1950-1999)" TO "from satellite images to black and white aerial photographs of the 80s decade (1980–1990) ". In table 2, authors should consider adding measurement units (meters) to their results. We agree with the reviewer, the caption of Table 2 changes FROM "Table 2. Spatial accuracy of the geometric-corrected image" TO "Table 2. Spatial accuracy of the geometric-corrected image (measurement units: meters)". In Figures 6 and 7 caption " rectangular rectangle." = " rectangle" We agree with the reviewer, in the caption of Figures 6 and 7 the text changed FROM "rectangular rectangle" TO "rectangle". In Table 3 for Landsat data fusion results in Green column and Green Line "emph0.770" = "0.770". We agree with the reviewer, this is a typographical error, in Table 3 the value changed FROM "emph0.770" TO "0.770". In the 3.4 Classification subsection, the authors' stating that they used 35 classification classes that subsequently are grouped into two classes. Which are these classes, and how they are grouped? These 35 classes are, for example, the classes that used are part of the 44 Corine Classes? An Unsupervised classification was undertaken, during which one does not know the classes in advance and does not choose the training fields of the algorithm to use, but sets the estimated number of classes, according to the number of coverages he / she believes to be present in the image (e.g. 10 different types of building cover, 10 different types of crops, 5 different types of natural vegetation, 5 different types of urban vegetation, 5 different types of roads, 10 different types of open surfaces, etc.). To make it more comprehensible the sentence changed FROM "35 classes of classification were selected, ..." TO "35 classes (by estimation) of classification were selected, ...". Can this methodology be applied to currently available data (new high-resolution aerial data and Sentinel imagery)? If the answer is positive, this should be stated in the manuscript as can be a crucial conclusion for applying this data fusion methodology, for example, to the automatic extraction of temporal information in Corine classification, reducing the photointerpretation process duration. The methodology proposes that there is a small difference (just a few days) between the date the b/w aerial photographs were taken and the MS satellite image, and that they are taken the same year and season. Today we do not take b/w aerial photographs and therefore the methodology cannot be applied to images of modern satellite systems. If there were modern b / w aerial photographs and with the MS Sentinel image they had the same date of capture, then the methodology could be applied.

Reviewer 2 Report

This paper proves experimentally the fusion of black and white aerial photographs with multispectral satellite images (i.e. Landsat-5).

I do not understand or identified what are the main differences, regarding the experiments, between the two use cases of Sparta and Pyrgos. Why was it necessary to present both? Please argument.

Unfortunately Introduction includes some related works as well. Would be useful to add in introduction the structure of the paper, how do you organise the paper. It is not clear what are the objectives of the paper and how do you reach the goal through a logical and systematical approach.

It is quite difficult to identify and understand the steps of the experimental approach, from the beginning of the paper. They are presented just they are, without any previous argumentation. Maybe at the beginning of section 3, the steps that follow in the paper could be enumerated or briefly described.

Give more explanation on computing and the meaning of the correlation given in Tables 3 and 4. How do you assess the processing by analysing the correlation?

It would be useful to further detail and explain the Section "3.4. Area classifications and measurements ”. How do you decide on the dimension of the geographical areas? Why do you select just 35 classes? What is the meaning of a class? Is it dependent on the visual characteristics only or is it important as well their meaning? All these operations are performed manually/assisted? Why the computation of areas is a good enough metrics? Is it a metrics or is it just one of the objectives/assessment of the experiments?

It would be useful to present an algorithmic method based on a metrics for deciding which photographs and satellite images could be combined. How the decision and the processing could be assisted or automatically achieved?

Some editing mistakes:
- in Fig 3, would be more useful to exemplify by the map including location of Sparta and Pyrgos.
- Change separator "/" by ";" in the text "e.g., Ikonos-2 at nadir 1 m PAN, 4 m MS/QuickBird-2 at nadir 0.65 m PAN, 2.6 m MS/WorldView-4 at nadir 0.31 m PAN, 1.24 m MS" -> "e.g., Ikonos-2 at nadir 1 m PAN, 4 m MS; QuickBird-2 at nadir 0.65 m PAN, 2.6 m MS; WorldView-4 at nadir 0.31 m PAN, 1.24 m MS"
- "not nessecarly at the same" -> "not necessarily at the same".
- make the correction on "emph0.770" in Table 3.
- make correction "rectangular rectangle" in Fig 6 and Fig 7.

Author Response

This paper proves experimentally the fusion of black and white aerial photographs with multispectral satellite images (i.e. Landsat-5). I do not understand or identified what are the main differences, regarding the experiments, between the two use cases of Sparta and Pyrgos. Why was it necessary to present both? Please argument. We would like to thank the reviewer for his constructive comments and suggestions that have improved our paper. We believe that a methodology should apply to more than one study area, and so we chose two random study areas (of random dimension) that are spatially distant from each other and that the pairs of the images (aerial and satellite image) of the data per region have different dates of acquisition. Unfortunately Introduction includes some related works as well. Would be useful to add in introduction the structure of the paper, how do you organise the paper. It is not clear what are the objectives of the paper and how do you reach the goal through a logical and systematical approach. In the introduction we analyze using the international bibliography, as we owe, the subject of study. This leaves a void in research, which, afterwards, we will try to cover by stating the aim of this paper: '' Therefore, the question refers to the possibility of re-using black and white aerial photographs after being improved by using spectral information. of archived satellite images. Obviously, it is important that the acquisition dates are identical or different for just a few days and that they are acquired in the same year. Thus, the fusion of these archived data and the results of their classifications are the objectives of this paper. It is quite difficult to identify and understand the steps of the experimental approach, from the beginning of the paper. They are presented just they are, without any previous argumentation. Maybe at the beginning of section 3, the steps that follow in the paper could be enumerated or briefly described. We agree with the reviewer and add a paragraph between Chapter 3 and subchapter 3.1. The paragraph is: After collecting the necessary data, the aerial triangulation-s of aerial photographs (for the production of orthophoto mosaics) and geometric correction-s of satellite images are conducted in the study area. Following are the image fusions of the orthophoto mosaics of the aerial photographs with the Landsat 5 orthoimages, as well as the checking of the quality (correlation tables) of the fused spectral information of the Multispectral (MS) satellite. Finally, the following classifications, both on the composite as well as on the MS ortho images, will be checked for their capability to provide accurate area measurements of both built and open surfaces. Give more explanation on computing and the meaning of the correlation given in Tables 3 and 4. How do you assess the processing by analysing the correlation? We agree with the reviewer and add text to subchapter 3.3. The change in the text is FROM ''..., producing a data-fusion image for each study area (Figures 6c, d and 7c, d). In order to create the correlation tables (Tables 43 and 54 [22–24]) of ortho multispectral… '' TO ''…, producing a data-fusion image for each study area (Figures 6c,d and 7c,d). The evaluation of fused image is based on qualitative-visual analysis and quantitative-statistical analysis. The qualitative-visual analysis is subjective and is directly related to the experience of the fused image creator (e.g. are more details recognized in the image or are colors, contrasts preserved? etc.) [7]. The quantitative-statistical analysis is objective and is based on spectral analysis of images. The most commonly used method is the correlation coefficient between the original bands of the MS image and the corresponding bands of the fused image. The correlation coefficient values range from -1 to 1. Usually, the values between the corresponding bands of the two images (MS and fused image) must be from 0.9 to 1, so that the fused image can be used for the e.g. successful classification of earth's surface coverings and objects [7, 22-24]. In order to create the correlation tables (Tables 43 and 54) of ortho multispectral…’’ In Chapter 4, to avoid confusion, we changed a sentence FROM ‘‘Consequently, composite images may be used to perform further classifications’’ TO ‘‘Consequently, despite the fact that the rates are not from 90-100% (ideally), these are generally satisfactory and composite images may be used to perform further classifications. '' It would be useful to further detail and explain the Section "3.4. Area classifications and measurements ”. How do you decide on the dimension of the geographical areas? Why do you select just 35 classes? What is the meaning of a class? Is it dependent on the visual characteristics only or is it important as well their meaning? All these operations are performed manually/assisted? Why the computation of areas is a good enough metrics? Is it a metrics or is it just one of the objectives/assessment of the experiments? We chose two random study areas (of random dimension) that are spatially distant from each other and that the pairs of the images (aerial and satellite image) of the data per region have different dates of acquisition. An Unsupervised classification was undertaken, during which one does not know the classes in advance and does not choose the training fields of the algorithm to use, but sets the estimated number of classes, according to the number of coverages he / she believes to be present in the image (e.g. 10 different types of building cover, 10 different types of crops, 5 different types of natural vegetation, 5 different types of urban vegetation, 5 different types of roads, 10 different types of open surfaces, etc.). To make it more comprehensible the sentence changed FROM "35 classes of classification were selected, ..." TO "35 classes (by estimation) of classification were selected, ...". It would be useful to present an algorithmic method based on a metrics for deciding which photographs and satellite images could be combined. How the decision and the processing could be assisted or automatically achieved? The solution is to use images with the same (or slightly different) date (day and year) of acquisition. On the other hand, we utilize images from different sensors, and so image fusion may not deliver the best results. Then, following the described methodological, the proportion of spectral information transmitted to the fused image must be determined. Some editing mistakes: - in Fig 3, would be more useful to exemplify by the map including location of Sparta and Pyrgos. We wanted to present the image with no additional information, and, therefore, we did not add North nor scale. We kindly ask that the image is kept as it is. - Change separator "/" by ";" in the text "e.g., Ikonos-2 at nadir 1 m PAN, 4 m MS/QuickBird-2 at nadir 0.65 m PAN, 2.6 m MS/WorldView-4 at nadir 0.31 m PAN, 1.24 m MS" -> "e.g., Ikonos-2 at nadir 1 m PAN, 4 m MS; QuickBird-2 at nadir 0.65 m PAN, 2.6 m MS; WorldView-4 at nadir 0.31 m PAN, 1.24 m MS" We agree with the reviewer, we changed the text FROM "e.g., Ikonos-2 at nadir 1 m PAN, 4 m MS/QuickBird-2 at nadir 0.65 m PAN, 2.6 m MS/WorldView-4 at nadir 0.31 m PAN, 1.24 m MS" TO "e.g., Ikonos-2 at nadir 1 m PAN, 4 m MS; QuickBird-2 at nadir 0.65 m PAN, 2.6 m MS; WorldView-4 at nadir 0.31 m PAN, 1.24 m MS ". - "not nessecarly at the same" -> "not necessarily at the same". We agree with the reviewer, we changed the text FROM "not nessecarly at the same" TO "not necessarily at the same". - make the correction on "emph0.770" in Table 3. It is a typographical mistake, we changed the error FROM "emph0.770" TO "0.770". - make correction "rectangular rectangle" in Fig 6 and Fig 7. We agree with the reviewer, in the caption of Figures 6 and 7 the text changed FROM "rectangular rectangle" TO "rectangle".

Round 2

Reviewer 1 Report

The authors responded to all my comments clearly, and they added the information needed. They improve the manuscript adding all the necessary content.

Reviewer 2 Report

The authors have improved the paper according to all the comments in the review.